# Warming proportional to cumulative carbon emissions not explained by heat and carbon sharing mixing processes

Nathan P. Gillett [1] ✉

The constant ratio of global warming to cumulative $CO_2$ emissions underpins the use of cumulative emissions budgets as policy tools, and the need to reach net zero $CO_2$ emissions to stabilize global mean temperature. Several studies have argued that this property arises because heat and carbon are mixed into the ocean by similar physical processes, and this argument was echoed in the latest Intergovernmental Panel on Climate Change report. Here we show that, contrary to this hypothesis, atmosphere-ocean fluxes of heat and carbon evolve very differently to each other in abrupt $CO_2$ increase experiments in five earth system models, and that changes in the atmosphere, ocean and land carbon pools all contribute to making warming proportional to cumulative emissions. Our results strongly suggest that this proportionality is not amenable to a simple physical explanation, but rather arises because of the complex interplay of multiple physical and biogeochemical processes.

Climate models consistently show that global mean warming is closely proportional to cumulative $CO_2$ emissions, with little sensitivity to emissions pathway[1-4]. This result featured prominently in the Intergovernmental Panel on Climate Change (IPCC) Fifth and Sixth Assessment Reports (AR5 and AR6), and leads directly to the concept of cumulative emissions budgets for $CO_2$, to which emissions must be limited in order to avoid exceeding particular temperature thresholds[1,3-5]. Such cumulative emissions budgets have been widely proposed as a policy tool to avoid exceeding warming thresholds, such as the 1.5 °C and 2 °C warming thresholds which are central to the Paris Agreement[3-7].

While the close proportionality of warming to cumulative emissions is a consistent property of earth system models (ESMs), the physical reasons for this proportionality are subject to debate. On multi-centennial timescales, the proportionality has been explained based on an exponentially increasing quasi-equilibrium airborne fraction of cumulative $CO_2$ emissions associated with ocean carbonate chemistry balancing a logarithmic dependence of radiative forcing on the $CO_2$ concentration increase[8,9]. Here we focus on the proportionality on decadal to centennial timescales, which requires two distinct properties of the climate system[3,10]. The first is that the global mean temperature response to a pulse emission of $CO_2$ is independent of the

background emissions scenario[2,11]. To first order, this has been explained by the balance between an approximately logarithmic dependence of radiative forcing on change in atmospheric $CO_2$ concentration, leading to a smaller change in radiative forcing per unit change in atmospheric $CO_2$ concentration at higher ambient $CO_2$, balanced by an increasing airborne fraction of emitted $CO_2$ at higher ambient $CO_2$, owing to a saturation of $CO_2$ sinks[2,8,9,11,12]. This property is a necessary but not sufficient condition for proportionality of warming to cumulative emissions across timescales, but we can use it to generalise results from one type of scenario to all others.

If we represent the cumulative emissions associated with a pulse emission as a step function $H(t)$, we can represent the cumulative emissions in a simulation in which the $CO_2$ concentration is instantaneously quadrupled[2,13,14], $E(t)$, as a weighted integral of pulse emission profiles (as illustrated schematically in Fig. 1a):

$$E(t) = \int_0^t w_1(t')H(t - t')dt' \tag{1}$$

If the temperature response to a pulse emission, $R(t)$, is independent of the background emissions scenario[11], it follows that the

[1]Canadian Centre for Climate Modelling and Analysis, Environment and Climate Change Canada, Victoria, BC, Canada. ✉e-mail: nathan.gillett@ec.gc.ca

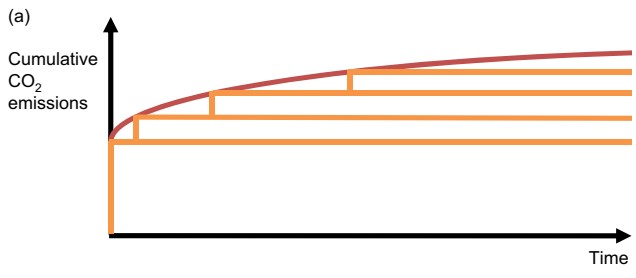

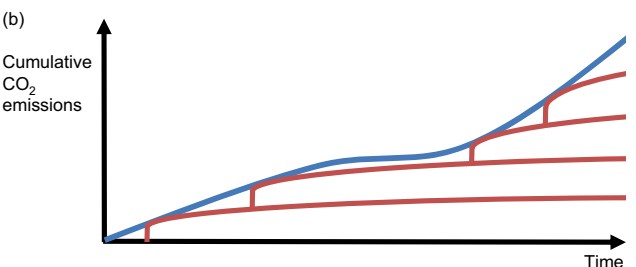

**Fig. 1 | Schematic illustrating how proportionality of warming to cumulative $CO_2$ emissions under an abrupt $4 \times CO_2$ scenario implies proportionality of warming to emissions under any arbitrary scenario. a** Cumulative emissions in a scenario in which atmospheric $CO_2$ concentration is abruptly increased and then held constant (red line), can be represented as a linear sum of pulse emission scenarios (orange lines; Eq. (1)). Since Caldeira and Kasting[11] demonstrated that the temperature response to a pulse emission of $CO_2$ is independent of the background emissions scenario, it follows that the temperature responses to a set of pulse emissions scenarios will add linearly (Eq. (2)), and also, based on (**a**), that the temperature response to a set of abrupt $CO_2$ increase scenarios will add linearly (Eq. (5)). **b** An arbitrary emissions scenario (blue line) can be represented as a linear sum of abrupt $CO_2$ increase scenarios (red lines; Eq. (3)). Hence if warming is proportional to cumulative emissions under an abrupt $CO_2$ increase scenario (such as abrupt $4 \times CO_2$), warming will be proportional to cumulative emissions under any arbitrary scenario.

temperature response to a set of pulse emission profiles, will be a linear sum of the temperature responses to the individual pulses. Hence the temperature response to an abrupt $4 \times CO_2$ simulation can be expressed as a weighted integral of the temperature response to individual pulse experiments, using the same weights as in Eq. (1):

$$\Delta T(t) = \int_0^t w_1(t') R(t - t') dt' \qquad (2)$$

Similarly, we can represent an arbitrary cumulative emissions profile, $E_A(t)$, as a weighted integral of $4 \times CO_2$ emissions profiles (shown schematically in Fig. 1b), and each $4 \times CO_2$ emissions profile can in turn can be represented as a weighted integral of pulse emissions profiles (substituting from Eq. (1)):

$$
\begin{aligned}
E_A(t) &= \int_0^t w_2(t'') E(t - t'') dt'' \\
&= \int_0^t w_2(t'') \int_0^{t-t''} w_1(t') H(t - t' - t'') dt' dt''
\end{aligned}
\qquad (3)
$$

Because the temperature response to a pulse emission is independent of the background emissions scenario[11] it follows that the temperature response to this arbitrary emissions profile, $\Delta T_A(t)$, can be written using the same weights as in Eq. (3), as:

$$\Delta T_A(t) = \int_0^t w_2(t'') \int_0^{t-t''} w_1(t') R(t - t' - t'') dt' dt'' \qquad (4)$$

and hence from Eq. (2):

$$\Delta T_A(t) = \int_0^t w_2(t'') \Delta T(t - t'') dt'' \qquad (5)$$

(for full generality this also requires that the temperature response to a negative pulse emission is also independent of the background emissions scenario). Comparing Eqs. (3) and (5), it is clear that if $\Delta T(t) \propto E(t)$ then $\Delta T_A(t) \propto E_A(t)$, i.e. if warming is proportional to cumulative emissions in a $4 \times CO_2$ experiment then warming is also proportional to cumulative emissions under any arbitrary scenario. Hence the second property of the climate system needed for proportionality of warming to emissions in general, is warming proportional to cumulative emissions in a $4 \times CO_2$ experiment. While we could have picked any scenario, we focus on understanding the proportionality in a $4 \times CO_2$ scenario because atmospheric $CO_2$ and radiative forcing are held constant throughout the experiment, making the climate response easier to represent analytically.

What is required for global mean temperature to be proportional to cumulative $CO_2$ emissions in an abrupt $4 \times CO_2$ experiment? Under such a scenario the surface temperature is clearly influenced by ocean heat uptake, and the cumulative $CO_2$ emissions, which may be diagnosed by summing the cumulative increases in atmospheric, land and ocean carbon, are clearly influenced by ocean carbon uptake. Solomon et al. [15] argue that the proportionality of warming to cumulative emissions following a complete cessation of emissions 'arises because long-term carbon dioxide removal and ocean heat uptake are both dependent on the same physics of deep-ocean mixing', and subsequent studies provide a similar explanation[2,16]. Bronselaer and Zanna[17] show that in transient simulations and observations the increase in ocean heat content and carbon content are approximately proportional to one another, and suggest that this is linked to the proportionality of warming to cumulative emissions. This argument is also reflected in the IPCC AR6[4] which assesses that 'the near-linear relationship between cumulative $CO_2$ emissions and global warming (TCRE) is thought to arise, to a large extent, from the compensation between the decreasing ability of the ocean to take up heat and $CO_2$ at higher cumulative $CO_2$ emissions, pointing to similar processes that determine ocean uptake of heat and carbon' and that 'a combination of unique chemical properties of seawater carbonate combined with shared physical ocean processes explain the coherence and scaling in the uptake and storage of both $CO_2$ and heat in the ocean..... [and] help understand the quasi-linear and path independence of properties of TCRE', where TCRE is the Transient Climate Response to Emissions, defined as the ratio of global warming to cumulative $CO_2$ emissions. However, the hypothesis that shared physical processes driving mixing of heat and carbon into the ocean drive the proportionality of warming to emissions has never been demonstrated analytically or in model experiments. Is this hypothesis correct? More specifically, if atmosphere-ocean fluxes of carbon and heat are proportional to each other does this imply a constant ratio of warming to cumulative emissions? Here, we examine evidence from an analytical model and from ESMs and find that this is not the case.

## Results
### Analytical model
We consider the response to an experiment in which $CO_2$ concentration is instantaneously quadrupled at time $t = 0$. Under these conditions, we can approximate the balance of heat fluxes at the surface by:

$$F_{4 \times CO_2} = q(t) + \lambda \Delta T(t) \qquad (6)$$

where $F_{4 \times CO_2}$ is the radiative forcing due to a quadrupling of $CO_2$, $q(t)$ is the global-average heat flux into the ocean (we neglect heat flux into the land), $\lambda$ is the climate feedback parameter, assumed constant, and

$\Delta T(t)$ is the global mean near-surface air temperature anomaly[18]. Rearranging:

$$\Delta T(t) = \frac{1}{\lambda}\left[F_{4 \times CO_2} - q(t)\right] \tag{7}$$

Since the total amount of carbon in the earth system is conserved, we can write:

$$E(t) = \Delta C_A + \Delta C_L(t) + \int_{t'=0}^{t'=t} f(t')dt' \tag{8}$$

where $E(t)$ is the cumulative $CO_2$ emissions expressed in PgC, $\Delta C_A$ is the change in atmospheric $CO_2$ in PgC, which is fixed in the experiment considered, $\Delta C_L(t)$ is the terrestrial carbon anomaly in PgC, $f(t)$ is the globally-integrated atmosphere-ocean flux of $CO_2$ in PgC/s, and $t'$ represents time in the integral.

Returning to our original question, if atmosphere-ocean fluxes of carbon and heat are proportional, does this imply that warming is proportional to cumulative emissions[15]? In terms of our equations, if $q(t) \propto f(t)$, does this imply that $\Delta T(t) \propto E(t)$? Examining Eqs. (7) and (8), it is clear that this is not true in general. Equation (8) contains $\Delta C_L(t)$ which is clearly unrelated to the atmosphere-ocean carbon flux, and moreover, while Eq. (7) contains atmosphere-ocean heat flux itself, Eq. (8) contains the integral of atmosphere-ocean carbon flux. What additional assumptions would be required for the Solomon et al.[15] hypothesis to be valid?

To address this question we take the approach of assuming that $q(t) \propto f(t)$ and $\Delta T(t) \propto E(t)$ and examining what this means for our analytical model. Hence we assume that $f(t) = \frac{f_0}{F_{4 \times CO_2}}q(t)$, where $f_0$ is the atmosphere-ocean carbon flux at $t = 0$, and noting that since $\Delta T(0) = 0$, then from Eq. (7) $q(0) = F_{4 \times CO2}$. If we further assume that $\Delta T(t) = \Lambda E(t)$, where $\Lambda$ is a constant TCRE, then from Eqs. (7) and (8) we obtain:

$$F_{4 \times CO_2} - q(t) = \lambda\Lambda\left[\Delta C_A + \Delta C_L(t) + \frac{f_0}{F_{4 \times CO2}}\int_{t'=0}^{t'=t} q(t')dt'\right] \tag{9}$$

and differentiating we obtain:

$$\frac{dq(t)}{dt} = -\lambda\Lambda\left[\frac{d\Delta C_L(t)}{dt} + \frac{f_0 q(t)}{F_{4 \times CO2}}\right] \tag{10}$$

Mathematically, an atmosphere-land carbon flux with a component proportional to the atmosphere-ocean carbon flux and a constant component is a possible solution of Eq. (10). However, on long timescales in a $4 \times CO_2$ experiment as the system approaches equilibrium the fluxes must approach zero; therefore the constant component must be zero. Further, the atmosphere-ocean carbon flux does not influence the atmosphere-land carbon flux at all in the experiment considered, since the atmospheric concentration of $CO_2$ is constant, hence there is no physical reason why the atmosphere-land carbon flux should be proportional to the atmosphere-ocean carbon flux. Hence, if the Solomon et al.[15] hypothesis is valid, and the proportionality between warming and cumulative emissions is driven by the proportionality between heat and carbon fluxes into the ocean, then land uptake of carbon must be negligible. Hence we assume that $\Delta C_L(t) = 0$, and we can solve Eq. (10) to obtain:

$$q(t) = F_{4 \times CO_2}e^{-\frac{\lambda\Lambda f_0}{F_{4 \times CO2}}t} \tag{11}$$

and hence:

$$f(t) = f_0 e^{-\frac{\lambda\Lambda f_0}{F_{4 \times CO2}}t} \tag{12}$$

Substituting into Eq. (9) we obtain:

$$\Delta C_A = 0 \tag{13}$$

Physically, such an exponentially decaying atmosphere-ocean carbon flux in the abrupt $4 \times CO_2$ experiment would be predicted by a model in which carbon is transferred from the atmosphere to a well-mixed ocean layer at a rate proportional to the difference in atmosphere and ocean carbon dioxide concentrations, with ocean carbon pool changes dominant and negligible changes in atmosphere and land carbon pools. Such an assumption would be consistent with the IPCC assessment[4] that 'the land carbon sink does not appear to play an important role in determining the linearity and path-independence of TCRE'. Similarly, such an exponentially decaying heat flux into the ocean in the abrupt $4 \times CO_2$ experiment would be predicted by a single layer energy balance model with heat capacity $\frac{F_{4 \times CO_2}}{\Lambda f_0}$. Substituting Eq. (11) into (7), and (12) and (13) into (8), we obtain:

$$\Delta T(t) = \frac{F_{4 \times CO_2}}{\lambda}\left[1 - e^{-\frac{\lambda\Lambda f_0}{F_{4 \times CO2}}t}\right] \tag{14}$$

$$E(t) = \frac{F_{4 \times CO_2}}{\Lambda\lambda}\left[1 - e^{-\frac{\lambda\Lambda f_0}{F_{4 \times CO2}}t}\right] \tag{15}$$

Hence atmosphere-ocean heat flux proportional to atmosphere-ocean $CO_2$ flux and warming proportional to cumulative emissions would only be fully realised in a single mixed-layer model in which changes in the ocean carbon pool dominated and changes in atmospheric and land carbon pools were negligible. How realistic is this? We answer this question by evaluating how well Eqs. (11)–(15) represent the climate system response to an abrupt increase in $CO_2$ concentration in five ESMs.

### Earth system model results

Figure 2a, d, g, j, m shows the evolution of the global mean near-surface air temperature anomaly in abrupt $4 \times CO_2$ simulations from five ESM simulations (black) from phase 6 of the Coupled Model Intercomparison Project (CMIP6[14]), compared to analytical model fits (grey) (see Methods). The analytical model underestimates the warming in the first 20–60 years of each simulation, and over-estimates it after this. Comparison of the atmosphere-ocean heat flux predicted by the analytical model with that simulated by the ESMs (Fig. 3a, d, g, j, m) indicates that the analytical model substantially overestimates the atmosphere-ocean heat flux for about the first 50 years, and then underestimates it in subsequent years, with the flux in the analytical model tending to zero much more rapidly than in the ESMs. This behaviour suggests that in later years of the experiment heat is taken up more rapidly by the ocean in the ESMs than a simple mixed layer model would suggest: This could for example be due to heat uptake by the deep ocean, including uptake driven by processes such as overturning and deep water formation. The higher ocean heat uptake in the ESMs compared to the analytical model after about year 50 explains their reduced warming in the latter part of the experiment compared to the analytical model (Fig. 2a, d, g, j, m).

Next we turn our attention to a comparison of carbon uptake and diagnosed emissions in the analytical model compared to the ESMs. Figure 2b, e, h, k, n, shows simulated anomalies in the atmosphere, ocean and land carbon pools in the $4 \times CO_2$ experiments of the ESMs, which sum to give the diagnosed cumulative $CO_2$ emissions. First we notice that the requirement that cumulative emissions are dominated by changes in the ocean carbon pool, and that atmospheric and land carbon changes are negligible, which follows from assuming that

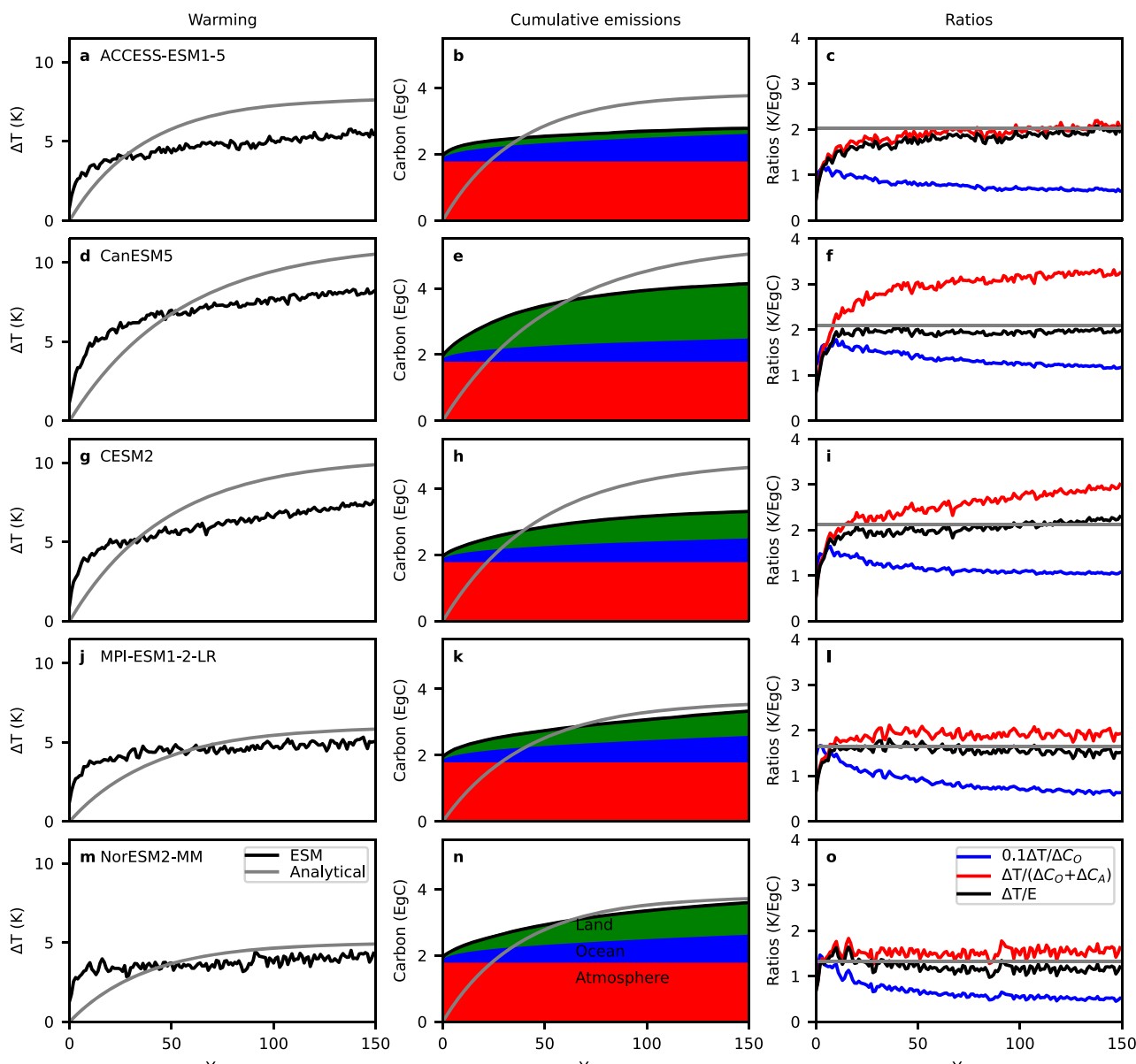

**Fig. 2 | Comparison of warming, cumulative carbon emissions and their ratio in abrupt 4 × CO₂ simulations from five Earth System Models (ESMs).** **a**, **d**, **g**, **j**, **m** Global mean near-surface air temperature anomalies relative to the preindustrial control ($\Delta T(t)$) in black, with corresponding analytical model fits in grey. **b**, **e**, **h**, **k**, **n** Changes in atmosphere (red), ocean (blue) and land (green) carbon pools, in EgC, and their sum (black line), which is equal to diagnosed cumulative CO₂ emissions. The grey line shows the cumulative CO₂ emissions in the analytical model. **c**, **f**, **i**, **l**, **o** The ratio in K/EgC of the warming to the increase in ocean carbon (scaled by a factor of 0.1 for display purposes; blue), the ratio of the warming to the increase in ocean plus atmosphere carbon (red), and the ratio of the warming to cumulative CO₂ emissions (i.e. the increase in ocean plus atmosphere plus land carbon; black). The grey line shows the models' Transient Climate Response to Emissions (TCRE), which is by construction equal to the ratio of warming to cumulative emissions in the analytical model. Each row of panels shows results from one ESM, which is named in the corresponding left panel. Source data are provided as a Source Data file.

warming is proportional to cumulative carbon emissions because atmosphere-ocean carbon and heat fluxes are proportional, is not at all consistent with the ESM simulations. In all cases, the change in the atmospheric carbon pool makes the largest contribution to diagnosed cumulative emissions, and in three of five models the change in the land carbon pool at the end of the simulation is larger than the change in the ocean carbon pool. Based on these results alone it is clear that any explanation for the proportionality of warming to cumulative carbon emissions which disregards the influence of land carbon uptake[4], or the relative changes in the atmosphere, ocean and land pools, is not a complete explanation. Nonetheless, we continue to evaluate the realism of the analytical model in other respects. The analytical model, which matches each ESM's TCRE by construction,

hence dramatically overestimates the ocean carbon uptake in each model by a factor of more than four over most of the simulation to compensate. Examination of the atmosphere-ocean carbon flux, shown in Fig. 2b, e, h, k, n, demonstrates that while the analytical model matches each ESM in the first year by construction, atmosphere-ocean carbon flux drops much more rapidly in the ESMs than in the analytical model. This is consistent with the well-known behaviour of ocean carbon in response to a pulse emission of CO₂, with some fraction taken up very rapidly by the surface ocean, and another fraction only taken up over centuries to millennia as the carbon is mixed deeper into the ocean[19]. For this reason, the carbon cycle response to a pulse emission is typically represented by a sum of multiple exponential terms with different timescales[19,20]. This is

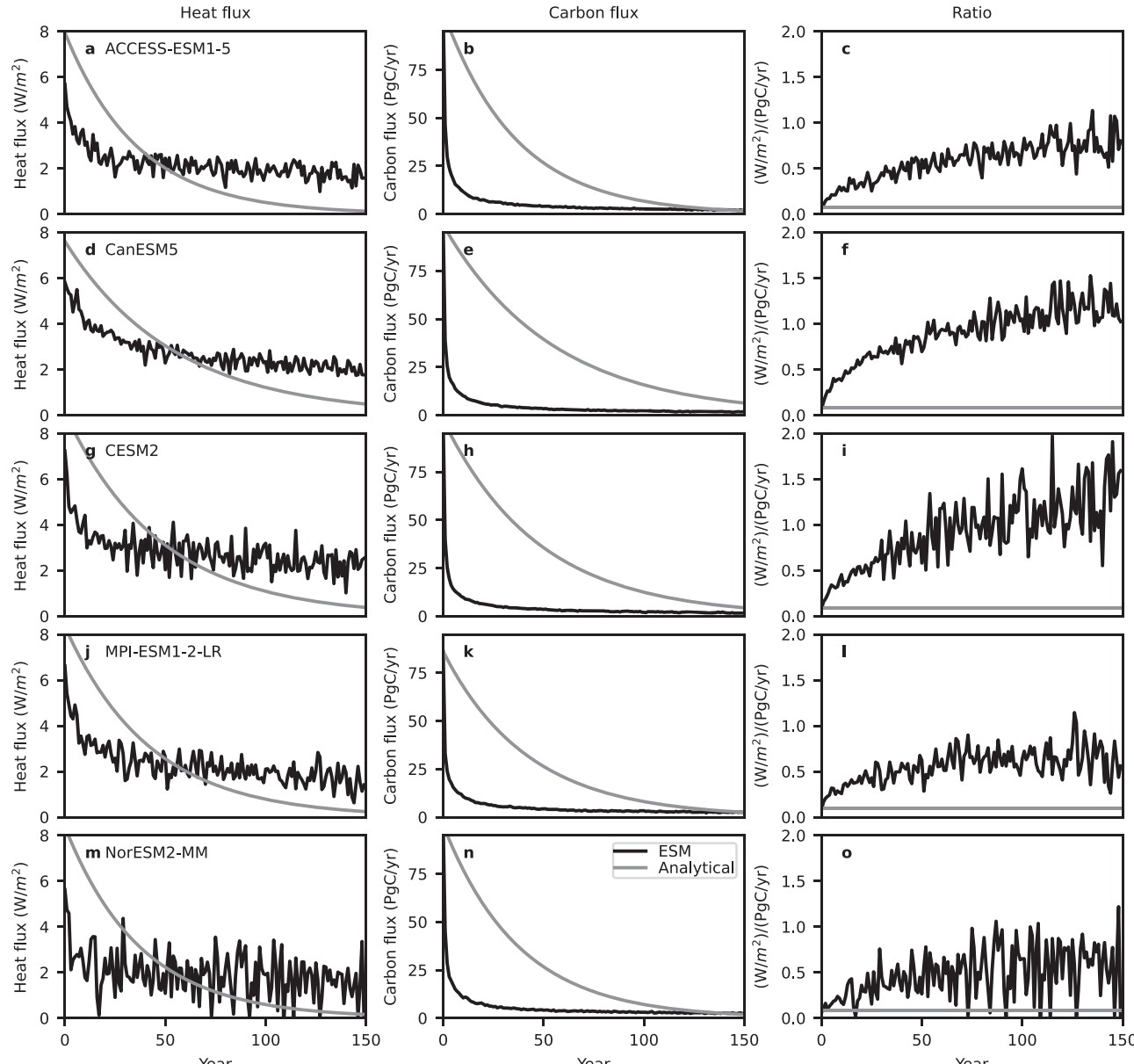

**Fig. 3 | Atmosphere-ocean heat flux, atmosphere-ocean carbon flux and the ratio of heat to carbon flux in abrupt 4 × CO₂ simulations from five Earth System Models (ESMs). a, d, g, j, m** Global mean atmosphere-ocean heat flux anomalies relative to preindustrial control. **b, e, h, k, n** Global total atmosphere-ocean carbon flux anomalies relative to preindustrial control. **c, f, i, l, o** The ratio of heat flux to carbon flux anomalies. In each case, black lines show ESM results, and grey lines show analytical model results. Each row of panels shows results from one ESM, which is named in the corresponding left panel. Source data are provided as a Source Data file.

another aspect of the carbon cycle response which is poorly represented by the analytical model.

As well as comparing the heat and carbon fluxes in the ESMs with the analytical model, we can also examine the assumption that ocean heat and carbon fluxes are proportional to each other, based on the hypothesis that warming is proportional to emissions because heat and carbon are mixed into the ocean by similar processe[15]. Figure 3c, f, i, l, o shows the ratio of heat to carbon fluxes into the ocean in each ESM. This ratio increases strongly throughout the experiment in three ESMs (ACCESS-ESM1-5, CanESM5 and CESM2), while in two ESMs (MPI-ESM1-2-LR and NorESM2-MM) it increases strongly in the first half of the experiment and stabilises in the second half. Thus it is far from true that the fluxes of heat and carbon into the ocean are proportional in this experiment in any of the five ESMs, and by inference, these fluxes are not driven by the same physical processes. Despite this, the

ratio of warming to cumulative emissions in these simulations is approximately constant from around year 10 of the simulations onwards (Fig. 2c, f, i, l, o, black line), albeit with weak increasing trends in two models (ACCESS-ESM1-5 and CESM2) and a decreasing trend in one (NorESM2-MM). Hence, the hypothesis that proportional heat and carbon fluxes explain the proportionality of warming to cumulative emissions in this experiment is not valid in these ESMs.

As well as the ratio of warming to diagnosed cumulative carbon emissions, Fig. 2c, f, i, l, o, also shows the ratio of warming to the ocean carbon pool anomaly (blue line). Since we are examining simulations with prescribed fixed atmospheric $CO_2$ concentration, the atmosphere, land, and ocean carbon pools do not interact, and we can interpret this as the ratio of warming to diagnosed cumulative emissions under the assumption that changes in land and atmosphere carbon are negligible. As shown in the figure, the ratio of warming to

the ocean carbon anomaly decreases throughout most of the experiment in all models and is much less constant over time than the ratio of warming to cumulative emissions. The behaviour of the ratio of warming to atmosphere plus ocean carbon anomalies (Fig. 2c, f, i, l, o, red line), interpretable as the ratio of warming to cumulative emissions under the assumption of negligible land uptake, varies between models, but in two models (CanESM5 and CESM2) it increases progressively through the simulation, and in general it is less constant in time than the ratio of warming to diagnosed cumulative carbon emissions, including land carbon. This analysis demonstrates that in all five models, the relative sizes of changes in the atmosphere and ocean carbon pools play an important role in the approximately constant ratio of warming to cumulative emissions, and in two of the models (CanESM5 and CESM2) changes in the land carbon pool also play an important role in this constancy, contrary to the IPCC AR6 assessment[4].

Since ref. 15 originally focused on explaining the proportionality of warming to emissions following a complete cessation of $CO_2$ emissions, and the $4 \times CO_2$ simulation contains a very abrupt change in forcing and might be considered an overly-exacting test of their hypothesis, we also examine a simulation of the climate and carbon response following a complete cessation of emissions. Supplementary Figs. 1 and 2 show the simulated response in a simulation in which atmospheric $CO_2$ concentration increases at 1% per year, followed by a complete cessation of emissions when cumulative emissions reach 1000 PgC[21], from CanESM5. Our analytical model assumes a constant $CO_2$ concentration, and therefore cannot be fitted to these simulations, but we can evaluate the ESM simulation to see if it is consistent with the hypothesis that warming is proportional to cumulative emissions because heat and carbon are mixed into the ocean by similar processes[15]. First, as in the $4 \times CO_2$ simulations, ocean carbon uptake makes only a small contribution to the diagnosed cumulative emissions (Supplementary Fig. 1b), with land uptake dominant, inconsistent with the Solomon et al.[15] hypothesis. Second, while the ratio of warming to cumulative emissions is almost constant, the ratio of warming to cumulative ocean carbon uptake declines through the simulation, while the ratio of warming to the sum of ocean and atmosphere carbon changes increases. Again, the constant ratio of warming to cumulative emissions can only be explained by considering the land carbon contribution to diagnosed cumulative emissions. Thirdly, even in this experiment without abrupt changes in $CO_2$ concentration, the ratio of atmosphere-ocean heat flux to atmosphere-ocean carbon flux is far from constant: It increases progressively during the period of increasing $CO_2$ concentration, before increasing more rapidly for about 30 years following the cessation of emissions. Overall, this analysis shows that it is not only in the $4 \times CO_2$ simulations that the hypothesis that proportional atmosphere-ocean heat and carbon fluxes explains the proportionality of warming to emissions fails, but that this hypothesis also fails to explain the behaviour seen in the zero emissions simulation as well.

## Discussion

Solomon et al.[15] argued that global warming is proportional to cumulative carbon emissions because long-term heat and carbon removal by the ocean are driven by the same physical processes, and a similar argument has been used to explain the proportionality across a range of timescales in many subsequent publications[2,16,17], including in the IPCC AR6[4]. Calderia and Kasting[11] demonstrated and explained why the temperature response to a pulse emission of $CO_2$ is independent of the background emissions scenario. This result implies that if warming is proportional to cumulative emissions in an abrupt $4 \times CO_2$ experiment, it will also be proportional under all other scenarios. The Solomon et al. [15] hypothesis implies that warming is proportional to cumulative carbon emissions because atmosphere-ocean heat and carbon fluxes are proportional. We demonstrate that this hypothesis

only provides a valid description of an abrupt $CO_2$ increase experiment under the assumption of a simple mixed-layer ocean which takes up heat and carbon, with negligible atmosphere and land carbon pool changes.

Turning to abrupt $4 \times CO_2$ simulations from five CMIP6 ESMs, we demonstrate that the analytical model implied by the hypothesis that warming is proportional to emissions because atmosphere-ocean heat and carbon fluxes are proportional is unrealistic in many ways. First, it provides a very poor fit to the atmosphere-ocean heat and carbon fluxes individually, and the ratio of atmosphere-ocean heat and carbon fluxes in the CMIP6 models is very far from constant, as this hypothesis implies. Second, ocean carbon changes make up only a relatively small part of the diagnosed cumulative emissions, with the relative contributions of atmosphere and ocean carbon pool changes playing an important role in driving the constancy of the ratio of warming to cumulative emissions in all models. Moreover, land carbon uptake also plays an important role in driving the constancy of the ratio of warming to cumulative emissions in some models.

How do we reconcile these results with those of previous studies? While Goodwin et al. [22] argue that 'the ocean sequestering of heat and carbon are both achieved in a similar manner', they do not demonstrate any proportionality of atmosphere-ocean heat and carbon fluxes, and, in their analytical model explaining the proportionality of warming to emissions, whereas they incorporate the instantaneous atmosphere-ocean heat uptake, they incorporate the integral of the atmosphere-ocean carbon uptake. Moreover, they have to assume that the ratio of land uptake to cumulative emissions is constant in time, which is not generally true in the simulations we examine (Fig. 2b, e, h, k, n). Williams et al. [23] expand on this framework, and demonstrate diagnostically how an approximately constant ratio of warming to emissions arises from the partial compensation of the effects of ocean heat and carbon uptake in an ESM, but they do not predict such constancy prognostically, and they also comment that 'there is no need for ocean sequestering of heat and carbon to always mirror each other'. MacDougall and Friedlingstein[12] and MacDougall[16] construct analytical models which are able to reproduce some aspects of the proportionality between warming and cumulative emissions under scenarios of increasing emissions. While MacDougall and Friedlingstein[12] argue that heat and carbon are taken up by different mechanisms, MacDougall[16] argues that heat and carbon are taken up by a similar mechanism, but shows differences in removal velocity for each, even in a scenario with linearly increasing emissions. Moreover, both studies are only able to reproduce the constant ratio of warming to emissions by assuming a constant ratio of land uptake to emissions. Bronselaer and Zanna[17] derive an analytical model in which cumulative ocean uptake of heat and carbon are proportional, but to do this they have to assume a constant rate of ocean heat uptake, and they also neglect land carbon uptake. Further, they only demonstrate this approximate proportionality in model simulations with progressively increasing $CO_2$ emissions, and not in abrupt $CO_2$ increase experiments, which represent a more exacting test of such a hypothesis. Overall, our results challenge the IPCC AR6[4] assessment that 'a combination of unique chemical properties of seawater carbonate combined with shared physical ocean processes explain the coherence and scaling in the uptake and storage of both $CO_2$ and heat in the ocean', because we find that heat and carbon uptake are not generally proportional, as this statement implies. We also find that these processes alone do not drive the proportionality of warming to emissions, as the assessment suggests. Further, our results for some models disagree with the IPCC AR6[4] assessment that 'The land carbon sink does not appear to play an important role in determining the linearity and path-independence of TCRE'.

Overall, on the decadal to centennial timescales considered here, we find that distinct properties of atmosphere-ocean heat and carbon fluxes, as well as the relative size of changes in the atmosphere and

ocean carbon pools, are important in driving the constancy of the ratio of warming to cumulative emissions, and in some models, land carbon uptake also plays an important role. Despite the political and scientific importance of this proportionality, our results strongly suggest that it is a chance result of many interacting physical and biogeochemical processes in the earth system, and not something which is amenable to simple physical explanation.

## Methods

We took monthly mean near-surface air temperature (CMIP6 variable name: tas), atmosphere-ocean heat flux (hfds), atmosphere-ocean $CO_2$ flux (fgco2), and net biosphere productivity (nbp), which is equal to atmosphere-land $CO_2$ flux, from the preindustrial control and abrupt $4 \times CO_2$ simulations from available CMIP6[14] ESM simulations. We required continuous output from the first 150 years of the abrupt $4 \times CO_2$ experiment for all four variables, ocean fraction (sftof), and all output on the native model grid, and we did not include more than one version of NCAR CESM2, because different versions share many model components. These criteria left us with output from the following five models: ACCESS-ESM1-5[24], CanESM5[25], CESM2[26], MPI-ESM1-2-LR[27] and NorESM2-MM[28]. After taking anomalies in abrupt $4 \times CO_2$ simulations relative to the preindustrial control (using the 150-year period starting at the branch time of abrupt $4 \times CO_2$, if provided), we calculated annual-mean global-mean near-surface air temperature and atmosphere-ocean heat flux anomalies, and annual-mean global total atmosphere-land and atmosphere-ocean carbon flux anomalies. Cumulative changes in land and ocean carbon pools were diagnosed from cumulative sums of atmosphere-land and atmosphere-ocean fluxes. The increase in the atmospheric carbon pool in the abrupt $4 \times CO_2$ experiment was taken as 1816 PgC in all models (corresponding to an assumed preindustrial $CO_2$ concentration of 284 ppm), since actual changes in specified $CO_2$ concentration were not available for all models. Cumulative $CO_2$ emissions were diagnosed by summing changes in the atmosphere, ocean and land carbon pools.

To compare the analytical model with the ESMs, we used Eqs. (11)–(15) to predict $q(t)$, $f(t)$, $\Delta T(t)$ and $E(t)$ for each model. We used diagnosed values of $F_{4 \times CO2}$ for each model[29], we calculated $\lambda$ from the equilibrium climate sensitivity of each model[30], and we calculated $f_O$ for each model as the annual mean of atmosphere-ocean carbon flux in the first year of its abrupt $4 \times CO_2$ experiment. We used reported values of TCRE for each model[31], which were calculated as the ratio of warming to cumulative $CO_2$ emissions at the time of $CO_2$ doubling in a simulation in which $CO_2$ increases at 1% per year.

## Data availability

All figures in this manuscript use CMIP6 data available here (https://esgf-node.llnl.gov/projects/cmip6/). The DOIs of the CMIP6 datasets used from each model are: ACCESS-ESM1-5: https://doi.org/10.22033/ESGF/CMIP6.2288, CanESM5: https://doi.org/10.22033/ESGF/CMIP6.1303, https://doi.org/10.22033/ESGF/CMIP6.1301, CESM2: https://doi.org/10.22033/ESGF/CMIP6.2185, MPI-ESM1-2-LR: https://doi.org/10.22033/ESGF/CMIP6.742, and NorESM2-MM: https://doi.org/10.22033/ESGF/CMIP6.506. Source data are provided with this paper.

## Code availability

The analysis code used in this study is based on ESMValTool and is available at https://github.com/ESMValGroup/ESMValTool/tree/gillett23.

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

## Acknowledgements

Thanks to Neil Swart for providing comments on the draft manuscript, and Liza Malinina for providing assistance with ESMValTool. I acknowledge the World Climate Research Programme, which, through its Working Group on Coupled Modelling, coordinated and promoted CMIP6. I thank the climate modelling groups for producing and making available their model output, the Earth System Grid Federation (ESGF) for archiving the data and providing access, and the multiple funding agencies who support CMIP6 and ESGF.

## Author contributions

N.P.G. carried out all the analysis in this paper and is the sole author of the paper.

## Competing interests

The author declares no competing interests.
