## [Peer Review File · Nature Communications]

REVIEWER COMMENTS

Reviewer #1 (Remarks to the Author):

Review of Gillet, N. "Is warming proportional to cumulative carbon emissions because heat and carbon are mixed into the ocean by the same process?"

This paper makes an important point that should be further appreciated in the literature: the approximate proportionality of the link between cumulative carbon emitted and global mean surface warming does not simply arise due to some similarity of processes involved with ocean heat and carbon uptake. Instead, it is the result of many interacting physical and biogeochemical factors.

The point is well demonstrated in the manuscript.

However, as the manuscript is written there are a couple of points that I believe require clarification, and so I suggest some minor revisions before publication.

Minor revision points:

1) The reasons why the cumulative emissions budgets are a useful policy tool.

In the second paragraph, the author states that the approximate proportionality of warming with cumulative carbon emitted leads directly to the concept of the cumulative carbon emissions budget as a policy tool to avoid exceeding warming thresholds.

There are two aspects for this proportionality: long timescale warming after emissions cease and transient warming during the period of carbon emission. In fact, both aspects are suggested by numerical models, but both are not required to make a cumulative emission budget a useful policy tool. Consider the two following statements:

(i) The warming on centennial timescales (after carbon emissions cease) is linearly related to cumulative carbon emitted – and this eventual centennial timescale warming is independent of the pathway by which the carbon was previously emitted.

(ii) The transient warming (during the period in which carbon is being emitted) is also approximately linearly related to the cumulative carbon emitted at any point in time, and is neither very much larger nor very much smaller than the centennial timescale warming (so that the Zero Emission Commitment is neither very large positive nor very large negative).

If only statement (i) were true, that is still enough to make future carbon emission budgets and net-zero emission rate targets both sensible policy goals. If we agree a warming target for the end of the century (to avoid the most dangerous climate change damages for year 2100), then we only need the centennial warming to be linearly related to cumulative carbon emitted to define out cumulative emission budget.

It is true that if the transient warming were very much larger than the centennial timescale warming (a large positive Zero Emission Commitment), then having a cumulative emissions budget as a policy goal would lead to temporary overshoot of the eventual warming target, but so long as the centennial timescale warming is approximately proportional to emissions and path independent then a cumulative emissions budget is still a sensible policy goal.

The second statement does make a cumulative emission budget even more useful, since it guarantees to have only a small temperature overshoot above the eventual year 2100 warming target. However, the second statement is not required to make the cumulative emissions budget a useful policy goal, but it does make the cumulative emission budget more useful in the short term.

This is important because of the manuscript and its conclusions relate to the second statement.

2) The reasons why warming is approximately linearly related to cumulative emissions in previous literature

The author states that the close proportionality between warming and cumulative emissions has been explained in the literature by:

“To first order, this has been explained by the balance between an approximately logarithmic dependence of radiative forcing on change in atmospheric CO₂ concentration, leading to a smaller change in radiative forcing per unit change in atmospheric CO₂ concentration at higher ambient CO₂, balanced by an increasing airborne fraction of emitted CO₂ at higher ambient CO₂, owing to a saturation of CO₂ sinks^{2,9,10.}”

The manuscript then considers how the reasons behind the increasing airborne fraction of emitted CO₂ at higher ambient CO₂ put forward in the literature (that the ocean heat and carbon uptake occurs in a similar way) are not adequate/too much of a simplification.

This statement quoted above (and the subsequent analysis) is accurate in regard to the literature relating to the transient warming only. Centennial warming has been shown to be approximately proportional to the cumulative carbon emitted because the approximate logarithmic increase in radiative forcing per unit increase in CO₂ is balanced by an approximate exponential increase in CO₂ per unit cumulative carbon emitted (see Goodwin et al., 2007 for the exponential relationship and Williams et al., 2012 for how this gives a linear centennial timescale warming per unit cumulative carbon emitted).

The reasons behind the exponential increase in CO₂ per unit cumulative carbon emitted on a centennial timescale are related to the carbonate chemistry of the ocean and the size of the ocean carbon pool (~38000PgC) in relation to the atmosphere (~500 PgC) (see Goodwin et al, 2007).

The reasons for this near linear warming from cumulative carbon emitted on centennial timescales (Williams et al., 2012; Goodwin et al., 2007) is well understood, and is enough to make a cumulative carbon emission budget a sensible policy goal to restrict warming in year 2100 to some agreed target.

What is less well explained and justified in the literature, is why the transient warming (during the period when emissions are ongoing) is also approximately linearly related to cumulative carbon emission. That is what this manuscript explores and explains well.

The author then later quotes Solomon et al., stating that “Solomon et al.¹³ argue that the proportionality of warming to emissions following a complete cessation of emissions ‘arises because long-term carbon dioxide removal and ocean heat uptake are both dependent on the same physics of deep-ocean mixing’, and subsequent studies provide a similar explanation^{2,14,15.}”

In fact, one of these references (Ref. 14: Goodwin et al., 2015) approaches things slightly differently to this statement, and is actually slightly more similar to this manuscript in purpose. Goodwin et al (2015) starts by taking the result that warming is approximately linearly related to cumulative carbon emissions on a centennial timescale (the findings that comes from Goodwin et al., 2007 and Williams et al., 2012) and then assesses why transient ocean uptake of heat and carbon makes this finding also apply (approximately) in the transient case considering RCP defined CO₂ concentration, and SRES defined emission, scenarios.

This manuscript does things differently by: (i) focussing on the 4xCO₂ situation, which tests the near-linear nature of the transient warming to the limit; and (ii) starting from the transient and working forwards (rather than starting from the centennial timescale solution and working backwards to the transient).

Summary:

I find the manuscript to show robust findings, that are of interest to readers from a wide variety of the climate science and policy literature.

However, I find that the current description of the place of the findings in that literature is a little off, and should be revised before publication.

Namely:

(1) It is previously well understood why the centennial timescale warming is linearly related to the cumulative carbon emitted, and that centennial near-linearity is enough to justify a cumulative carbon budget approach.

(2) This study provides new insight on why the transient warming is also nearly linearly related to cumulative carbon emitted (which reveals why the transient warming does not overshoot the eventual stabilised warming too far and makes the further case for cumulative carbon emission budgets as a policy tool).

References:

Goodwin, P., R.G. Williams, M.J. Follows, S. Dutkiewicz (2007), Ocean-atmosphere partitioning of anthropogenic carbon dioxide on centennial timescales, *Global Biogeochemical Cycles*, 21, GB1014, <https://doi.org/10.1029/2006GB002810>.

Goodwin, P., R.G. Williams and A. Ridgwell (2015), Sensitivity of climate to cumulative carbon emissions due to compensation of ocean heat and carbon uptake, *Nature Geoscience*, Vol. 8, p29-34. <https://doi.org/10.1038/ngeo2304>.

Williams, R. G., P. Goodwin, A. Ridgwell, and P. L. Woodworth (2012), How warming and steric sea level rise relate to cumulative carbon emissions, *Geophys. Res. Lett.*, 39, L19715, <https://doi.org/10.1029/2012GL052771>.

Reviewer #2 (Remarks to the Author):

Review of "Is warming proportional to cumulative carbon emissions because heat and carbon are mixed into the ocean by the same processes?" by Nathan P. Gillett

The author is scrutinizing the hypothesis that the approximate proportionality between cumulative carbon emissions and global warming (i.e., the constancy of the transient climate response to cumulative emissions, TCRE) can be explained (partly) by the fact that heat and carbon are taken up by the ocean in a similar manner. Specifically, the author is constructing a simple model of the climate carbon system, in which carbon and heat fluxes are proportional in a setting where atmospheric CO₂ is instantaneously quadrupled. In this simplified model, the temperature response is strictly proportional to cumulative emissions. Comparing the simple model results with 4xCO₂ simulations from complex Earth system models, the author finds significant deviations between ESMs and the simple model, which leads to the conclusion that the hypothesis that guided the construction of the simple model must have been wrong.

As the author points out, the statement that TCRE arises mainly due to ocean uptake of heat and carbon in an approximately fixed proportion has become commonplace over the past decade and revisiting and checking this assumption is a very useful exercise. Improving our understanding of the physical basis of TCRE (or understanding what is NOT the physical basis) would be a considerable

advancement. The text is generally well written and the method is original and (technically) sound. However, the author did not convince me that the main conclusion (TCRE is NOT due to a proportionality of heat and carbon fluxes) is a necessary (the only possible) consequence of the mismatch between simple and ESM results in the particular model experiment considered in this study. I have listed 3 main points, which I do not find convincing, below together with a few more minor comments.

Main points:

1) Choice of the 4xCO₂ experiment: The 4xCO₂ experiment is an idealized, quite extreme experiment where a large radiative forcing is introduced instantaneously. I am not convinced that this experiment design is the most well suited to study the proportionality of heat and carbon fluxes. For example, the initial CO₂ fluxes f_0 diagnosed from the ESMs are huge. Maybe it is not reasonable to expect a simple model to perform well in such a situation? Also the ESMs show a transient behavior of TCRE in this experiment for up to several decades (Fig. 2, 3rd column) before TCRE stabilizes. The 4xCO₂ experiment certainly doesn't represent a situation for which Solomon et al. made their claim about heat and CO₂ fluxes being mixed down by similar ocean processes (they consider the climate system after complete cessation of all emissions and on long time scales, ~1000 years). I am not saying that the 4xCO₂ experiment can't be used, but wouldn't it make more sense to choose the initial time for comparing the simple model and ESMs differently? What if the initial time is shifted, say by 30 years? For the ESMs it takes 1-4 decades until a stable TCRE is reached according to Fig 2. Also, this would lead to $\Delta C_a=0$, which otherwise is clearly an unreasonable assumption as pointed out by the author.

2) Functional form of the simple model: To what extent does the functional form for the simple model hinge on the choice of the 4xCO₂ experiment for studying TCRE? The exponential decay functions for $q(t)$ and $f(t)$ work only because the quadrupling of CO₂ results in a constant radiative forcing, and in CO₂ fluxes that are largest at initial time t_0 ? While I see that the simple model is a solution in which $q(t) \sim f(t)$ for the special case of the 4xCO₂ simulation, it remains unclear to me whether this is the only possible solution? Because of the decaying form of $f(t)$ and $q(t)$, this model can't be suitable to approximate heat and carbon fluxes under other trajectories of CO₂ concentration, for example under the 1% CO₂ experiment which is used to operationally define TCRE in ESMs?

3) Now, if I accept that it makes sense to compare the simple model as developed by the author to ESM results, I find the comparison is unnecessarily strict. All model parameters for the simple model seem to be taken from the respective ESM (according to the methods section). However, the purpose here is to test whether a model in which the proportionality of warming and cumulative

emissions follows from $q(t) \sim f(t)$ could reproduce (partly) ESM results. The purpose is not to test whether a single layer EBM and a very simple carbon cycle model with realistic parameters (=parameters corresponding to an ESM) can reproduce results from an ESM simulation (we know that it can't). So, it would be more appropriate to use one of the parameters to tune the simple model to the respective ESM. Since λ cancels out in the calculation of TCRE (eq. 9), this would be a possible choice. My suspicion is that through "tuning" of λ it will be possible to obtain a relatively good fit for ΔT and carbon fluxes for the last 100 years of the simulation in all ESMs (as is already the case for 2 of the models). If this would work, then the mismatch would come down to a question of timescales: For the short timescale the simple model doesn't explain ESM results, but for the longer it approximately can (this is already the case for MIP-ESM and NorESM2).

Other points:

In the 2nd paragraph of the text, the author writes: "As shown in Figure 1, this property implies that if warming is proportional to cumulative emissions under one type of emissions scenario, it will be proportional to warming in all others." In my opinion, Fig. 1 has little explanatory power. I would rather prefer some (mathematically based) inference, or a reference in which such can be found.

In the derivation of the simple model it is assumed early on that $\Delta C_I(t)=0$ (stating that this is done because IPCC has assessed that it is not important). But doesn't $\Delta C_I(t)=0$ also follow from the derivation of the model itself, if one imposes to have $\Delta T \sim E$ (same as for ΔC_a). So, wouldn't it make more sense to still have ΔC_I in eq. (8) and then state that it needs to be zero for deriving eq. (9)?

The author states several times that the results obtained with the simple model contradict the study of Solomon et al. (2009). It is worth noting, however, that Solomon et al. consider the climate system after complete cessation of all emissions and on long time scales (1000 years), something that is not at all represented by the model experiment investigated here. So, I would recommend a more careful wording here.

Response to reviewers' comments on "Is warming proportional to cumulative carbon emissions because heat and carbon share ocean mixing processes?"

Reviewer #1 (Remarks to the Author):

Review of Gillet, N. "Is warming proportional to cumulative carbon emissions because heat and carbon are mixed into the ocean by the same process?"

This paper makes an important point that should be further appreciated in the literature: the approximate proportionality of the link between cumulative carbon emitted and global mean surface warming does not simply arise due to some similarity of processes involved with ocean heat and carbon uptake. Instead, it is the result of many interacting physical and biogeochemical factors.

The point is well demonstrated in the manuscript.

However, as the manuscript is written there are a couple of points that I believe require clarification, and so I suggest some minor revisions before publication.

Thanks to the reviewer for the positive comments on the manuscript, and I'm pleased that he or she finds the conclusions to be well-supported. I have made the revisions requested by the reviewer, as described below.

Minor revision points:

1) The reasons why the cumulative emissions budgets are a useful policy tool.

In the second paragraph, the author states that the approximate proportionality of warming with cumulative carbon emitted leads directly to the concept of the cumulative carbon emissions budget as a policy tool to avoid exceeding warming thresholds.

There are two aspects for this proportionality: long timescale warming after emissions cease and transient warming during the period of carbon emission. In fact, both aspects are suggested by numerical models, but both are not required to make a cumulative emission budget a useful policy tool. Consider the two following statements:

(i) The warming on centennial timescales (after carbon emissions cease) is linearly related to cumulative carbon emitted – and this eventual centennial timescale warming is independent of the pathway by which the carbon was previously emitted.

(ii) The transient warming (during the period in which carbon is being emitted) is also approximately linearly related to the cumulative carbon emitted at any point in time, and is neither very much larger nor very much smaller than the centennial timescale warming (so that the Zero Emission Commitment is neither very large positive nor very large negative).

If only statement (i) were true, that is still enough to make future carbon emission budgets and net-zero emission rate targets both sensible policy goals. If we agree a warming target for the end of the century (to avoid the most dangerous climate change damages for year 2100), then we only need the centennial warming to be linearly related to cumulative carbon emitted to define out cumulative emission budget.

It is true that if the transient warming were very much larger than the centennial timescale warming (a large positive Zero Emission Commitment), then having a cumulative emissions budget as a policy goal would lead to temporary overshoot of the eventual warming target, but so long as the centennial timescale warming is approximately proportional to emissions and path independent then a cumulative emissions budget is still a sensible policy goal.

The second statement does make a cumulative emission budget even more useful, since it guarantees to have only a small temperature overshoot above the eventual year 2100 warming target. However, the second statement is not required to make the cumulative emissions budget a useful policy goal, but it does make the cumulative emission budget more useful in the short term.

This is important because of the manuscript and its conclusions relate to the second statement.

Thanks to the reviewer for the comment and for raising this issue. I agree that centennial-scale warming being proportional to cumulative carbon emissions could by itself make a cumulative carbon budget a useful policy tool to some extent, since it would lead to a particular level of cumulative emissions for a particular long-term temperature goal. However, as the reviewer points out, a transient warming linearly related to cumulative emissions does make the budget even more useful, because if the transient warming depended on the pathway and not only on cumulative emissions, then different pathways with the same cumulative emissions could result in different peak warming, and hence potentially different impacts and costs. Even if the transient warming were less than the centennial warming, this would still allow different pathways with the same cumulative emissions to result in different peak warming, since overshoot scenarios with negative emissions later this century would then result in lower peak warming than scenarios with the same cumulative emissions but without overshoot. Elsewhere the reviewer points to Williams et al. (2012) and Goodwin et al. (2007), which explain this proportionality on long timescales, and I was remis in not citing these papers in the original version of my manuscript. I have now added a sentence describing these results and citing these studies, prior to my discussion of the reasons for proportionality across a range of timescales.

On multi-centennial timescales, the proportionality has been explained based on an exponentially increasing quasi-equilibrium airborne fraction of cumulative CO₂ emissions associated with ocean carbonate chemistry balancing a logarithmic dependence of radiative forcing on the CO₂ concentration increase^{8,9}. Here we focus on the proportionality across a range of timescales, which requires two distinct properties of the climate system^{3,10}.

Note however that my study does adopt a slightly different framework from that proposed by the reviewer, starting from the Caldeira and Kasting (1993) result of the temperature response to a pulse emission being independent of the background scenario, and asking what else is needed to give warming proportional to cumulative emissions across all timescales, not just in the transient case. Thus I do now cite and discuss the Goodwin et al. (2007) and Williams et al. (2012) results, but I have not reworked the manuscript to only concentrate on the transient warming.

2) The reasons why warming is approximately linearly related to cumulative emissions in previous literature

The author states that the close proportionality between warming and cumulative emissions has been explained in the literature by:

“To first order, this has been explained by the balance between an approximately logarithmic dependence of radiative forcing on change in atmospheric CO₂ concentration, leading to a smaller change in radiative forcing per unit change in atmospheric CO₂ concentration at higher ambient CO₂, balanced by an increasing airborne fraction of emitted CO₂ at higher ambient CO₂, owing to a saturation of CO₂ sinks^{2,9,10}.”

The manuscript then considers how the reasons behind the increasing airborne fraction of emitted CO₂ at higher ambient CO₂ put forward in the literature (that the ocean heat and carbon uptake occurs in a similar way) are not adequate/too much of a simplification.

This statement quoted above (and the subsequent analysis) is accurate in regard to the literature relating to the transient warming only. Centennial warming has been shown to be approximately proportional to the cumulative carbon emitted because the approximate logarithmic increase in radiative forcing per unit increase in CO₂ is balanced by an approximate exponential increase in CO₂ per unit cumulative carbon emitted (see Goodwin et al., 2007 for the exponential relationship and Williams et al., 2012 for how this gives a linear centennial timescale warming per unit cumulative carbon emitted).

The reasons behind the exponential increase in CO₂ per unit cumulative carbon emitted on a centennial timescale are related to the carbonate chemistry of the ocean and the size of the ocean carbon pool (~38000PgC) in relation to the atmosphere (~500 PgC) (see Goodwin et al, 2007).

The reasons for this near linear warming from cumulative carbon emitted on centennial timescales (Williams et al., 2012; Goodwin et al., 2007) is well understood, and is enough to make a cumulative carbon emission budget a sensible policy goal to restrict warming in year 2100 to some agreed target.

Thanks to the reviewer for flagging these studies, and apologies for omitting references to these studies in the original version of my manuscript. As noted above, I have now added the following text to the introduction to describe the results of these studies and better set my study in context:

On multi-centennial timescales, the proportionality has been explained based on an exponentially increasing quasi-equilibrium airborne fraction of cumulative CO₂ emissions associated with ocean carbonate chemistry balancing a logarithmic dependence of radiative forcing on the CO₂ concentration increase^{8,9}. Here we focus on the proportionality across a range of timescales, which requires two distinct properties of the climate system^{3,10}.

I have also added a sentence describing the related study by Williams et al. (2015) to the discussion section. I acknowledge that a linear dependence of warming on cumulative emissions on centennial timescales would make a cumulative emissions budget a somewhat useful policy goal, but the

transient behaviour is also important in determining the peak level of warming in response to a given level of cumulative emissions. Thus the changes I have made on this point are limited to those just described.

What is less well explained and justified in the literature, is why the transient warming (during the period when emissions are ongoing) is also approximately linearly related to cumulative carbon emission. That is what this manuscript explores and explains well.

The author then later quotes Solomon et al., stating that “Solomon et al.¹³ argue that the proportionality of warming to emissions following a complete cessation of emissions ‘arises because long-term carbon dioxide removal and ocean heat uptake are both dependent on the same physics of deep-ocean mixing’, and subsequent studies provide a similar explanation^{2,14,15.}”

In fact, one of these references (Ref. 14: Goodwin et al., 2015) approaches things slightly differently to this statement, and is actually slightly more similar to this manuscript in purpose. Goodwin et al (2015) starts by taking the result that warming is approximately linearly related to cumulative carbon emissions on a centennial timescale (the findings that comes from Goodwin et al., 2007 and Williams et al., 2012) and then assesses why transient ocean uptake of heat and carbon makes this finding also apply (approximately) in the transient case considering RCP defined CO₂ concentration, and SRES defined emission, scenarios.

Thanks to the review for flagging this. Although Goodwin et al. (2015) do say the following “The ocean sequestering of heat and carbon are both achieved in a similar manner: there is a relatively rapid drawdown of heat and carbon from the atmosphere into the surface mixed layer on annual to decadal timescales (Fig. 4a); a subsequent ventilation of the main thermocline and upper ocean over decades to centuries (Fig. 4b); and a slower ventilation of the deep ocean over many centuries, or even millennia (Fig. 4c).”, they do not say that the proportionality of warming to cumulative emissions arises directly from this. Therefore I agree with the reviewer that my reference to Goodwin et al. (2015) in the sentence the reviewer cites above is misplaced, and I have removed it, and I have also removed it from the first sentence of the Discussion for the same reason. (Note that the results of Goodwin et al. (2015) are still discussed later in the manuscript).

This manuscript does things differently by: (i) focussing on the 4xCO₂ situation, which tests the near-linear nature of the transient warming to the limit; and (ii) starting from the transient and working forwards (rather than starting from the centennial timescale solution and working backwards to the transient).

Summary:

I find the manuscript to show robust findings, that are of interest to readers from a wide variety of the climate science and policy literature.

However, I find that the current description of the place of the findings in that literature is a little off, and should be revised before publication.

Namely:

(1) It is previously well understood why the centennial timescale warming is linearly related to the cumulative carbon emitted, and that centennial near-linearity is enough to justify a cumulative carbon budget approach.

(2) This study provides new insight on why the transient warming is also nearly linearly related to cumulative carbon emitted (which reveals why the transient warming does not overshoot the eventual stabilised warming too far and makes the further case for cumulative carbon emission budgets as a policy tool).

Thanks again for the positive comments. As described above I have now described and cited the results of Williams et al. (2012) and Goodwin et al. (2007), and removed the erroneous reference to Goodwin et al. (2015), as suggested.

References:

Goodwin, P., R.G. Williams, M.J. Follows, S. Dutkiewicz (2007), Ocean-atmosphere partitioning of anthropogenic carbon dioxide on centennial timescales, *Global Biogeochemical Cycles*, 21, GB1014, <https://doi.org/10.1029/2006GB002810>.

Goodwin, P., R.G. Williams and A. Ridgwell (2015), Sensitivity of climate to cumulative carbon emissions due to compensation of ocean heat and carbon uptake, *Nature Geoscience*, Vol. 8, p29-34. <https://doi.org/10.1038/ngeo2304>.

Williams, R. G., P. Goodwin, A. Ridgwell, and P. L. Woodworth (2012), How warming and steric sea level rise relate to cumulative carbon emissions, *Geophys. Res. Lett.*, 39, L19715, <https://doi.org/10.1029/2012GL052771>.

Reviewer #2 (Remarks to the Author):

Review of "Is warming proportional to cumulative carbon emissions because heat and carbon are mixed into the ocean by the same processes?" by Nathan P. Gillett

The author is scrutinizing the hypothesis that the approximate proportionality between cumulative carbon emissions and global warming (i.e., the constancy of the transient climate response to cumulative emissions, TCRE) can be explained (partly) by the fact that heat and carbon are taken up by the ocean in a similar manner. Specifically, the author is constructing a simple model of the climate carbon system, in which carbon and heat fluxes are proportional in a setting where atmospheric CO₂ is instantaneously quadrupled. In this simplified model, the temperature response is strictly proportional to cumulative emissions. Comparing the simple model results with 4xCO₂ simulations from complex Earth system models, the author finds significant deviations between ESMs and the simple model, which leads to the conclusion that the hypothesis that guided the construction of the simple model must have been wrong.

As the author points out, the statement that TCRE arises mainly due to ocean uptake of heat and carbon in an approximately fixed proportion has become commonplace over the past decade and revisiting and checking this assumption is a very useful exercise. Improving our understanding of the physical basis of TCRE (or understanding what is NOT the physical basis) would be a considerable advancement. The text is generally well written and the method is original and (technically) sound. However, the author did not convince me that the main conclusion (TCRE is NOT due to a proportionality of heat and carbon fluxes) is a necessary (the only possible) consequence of the mismatch between simple and ESM results in the particular model experiment considered in this study. I have listed 3 main points, which I do not find convincing, below together with a few more minor comments.

Thanks to the reviewer for the positive comments on the motivation for the manuscript, and I'm glad that he or she agrees that checking the widely-held belief that constant TCRE arises due to ocean uptake of heat and carbon being proportional is a useful exercise, and that the text is well written and the method is original and technically sound. I will address the reviewer's concerns over my refutation of this hypothesis in response to the individual comments below.

Main points:

1) Choice of the 4xCO₂ experiment: The 4xCO₂ experiment is an idealized, quite extreme experiment where a large radiative forcing is introduced instantaneously. I am not convinced that this experiment design is the most well suited to study the proportionality of heat and carbon fluxes. For example, the initial CO₂ fluxes f_0 diagnosed from the ESMs are huge. Maybe it is not reasonable to expect a simple model to perform well in such a situation? Also the ESMs show a transient behavior of TCRE in this experiment for up to several decades (Fig. 2, 3rd column) before TCRE stabilizes. The 4xCO₂ experiment certainly doesn't represent a situation for which Solomon et al. made their claim about heat and CO₂ fluxes being mixed down by similar ocean processes (they consider the climate system after complete cessation of all emissions and on long time scales, ~1000 years). I am not saying that the 4xCO₂

experiment can't be used, but wouldn't it make more sense to choose the initial time for comparing the simple model and ESMs differently? What if the initial time is shifted, say by 30 years? For the ESMs it takes 1-4 decades until a stable TCRE is reached according to Fig 2. Also, this would lead to $\Delta C_a=0$, which otherwise is clearly an unreasonable assumption as pointed out by the author.

Thanks to the reviewer for raising this point. In response to this comment, in particular the comment that the Solomon et al. hypothesis was formulated to explain the response in a simulation following a complete cessation of CO₂ emissions, and the suggestion from the editor that I explore other approaches than analysis of the 4xCO₂ experiment, I have carried out and added new analysis of a zero emissions simulation of CanESM5. (Note that of the ESMs used, the necessary variables were only published on ESGF for CanESM5 and MPI-ESM1-2-LR, and MPI-ESM1-2-LR lacked the required ocean grid cell area files for this simulation needed to process the data in ESMValTool. Therefore I was only able to carry out this analysis for CanESM5). These results are shown in two new figures (Supplementary Figures 1 and 2) and are discussed in a new paragraph at the end of the Earth System Model results section. These results allow me to test in the zero emissions simulation i) whether the cumulative emissions are dominated by ocean carbon uptake, as the Solomon et al. hypothesis requires, ii) whether or not land carbon uptake contributes to the constancy of the ratio of warming to cumulative emissions, and iii) whether ocean uptake of heat and carbon are proportional to each other in these simulations with smoothly varying CO₂ concentration. The analysis indicates that even in this zero emissions simulation, i) cumulative emissions are not dominated by ocean carbon uptake, and land carbon uptake is the biggest contributor to diagnosed cumulative emissions by the end of the simulation analysed, ii) the ratio of warming to ocean carbon uptake is not constant, but declines over time, whereas the ratio of warming to total cumulative emissions including land uptake is almost constant, and iii) the ratio of atmosphere-ocean heat and carbon fluxes varies over time through the simulations. Hence this analysis indicates that warming is not proportional to cumulative emissions because heat and carbon are proportional to one another in this simulation, consistent with results obtained for the 4xCO₂ simulations.

Constructing a simple model to fit the response beginning at year 30 after a quadrupling of CO₂ concentrations, but with warming proportional to cumulative emissions, and heat and carbon fluxes proportional, would not be straightforward. The main issue is that at year 30 of the experiment, the climate system is responding strongly to cumulative emissions of CO₂ emitted prior to this, which include the change in the atmosphere, land and ocean carbon inventories in the first 30 years of the experiment, and this is indeed the main contribution to the cumulative emissions through the whole experiment (and the main driver of the temperature response). Thus it is not possible to construct a physical model of the climate system response beginning at year 30 of the experiment, which ignores its behaviour in the first 30 years. The model would have to assume some level of cumulative emissions and warming in the prior 30 year period, and it is not clear what the physical basis for this would be. Moreover, Figure 3 demonstrates that in ESMs, the ratio of heat flux to carbon flux increases considerably even after year 30 of the experiment, while Figure 2 demonstrates that ocean carbon uptake is not dominant in this part of the experiment. Thus, even if a simple model could somehow be constructed for this part of the experiment only, the key assumptions underlying my simple model would still not be valid. Therefore I have relied on the new analysis of the zero emissions simulation to address this comment.

2) Functional form of the simple model: To what extent does the functional form for the simple model hinge on the choice of the 4xCO₂ experiment for studying TCRE? The exponential decay functions for $q(t)$ and $f(t)$ work only because the quadrupling of CO₂ results in a constant radiative forcing, and in CO₂ fluxes that are largest at initial time t_0 ? While I see that the simple model is a solution in which $q(t) \sim f(t)$ for the special case of the 4xCO₂ simulation, it remains unclear to me whether this is the only possible solution? Because of the decaying form of $f(t)$ and $q(t)$, this model can't be suitable to approximate heat and carbon fluxes under other trajectories of CO₂ concentration, for example under the 1% CO₂ experiment which is used to operationally define TCRE in ESMs?

To answer the reviewer's first question – yes, the reviewer is correct that the functional form of the simple model works because the radiative forcing and CO₂ concentration are constant in the 4xCO₂ experiment. The 1pctCO₂ experiment would have linearly increasing radiative forcing in equation 7, and exponentially increasing ΔC_A in equation 8, and an analytical solution with $q(t) \propto f(t)$ and $\Delta T(t) \propto E(t)$ would not be possible. However, as argued in the second and third paragraph of the introduction, based on the Caldeira and Kasting (1993) result, if warming is proportional to cumulative emissions in one kind of experiment, it will be proportional in all others (this discussion is now considerably expanded and includes a mathematical explanation). Thus if we can explain why warming is proportional to cumulative emissions in one scenario, we can explain it in all others. Hence I focus on the 4xCO₂ experiment. This reason for choosing the 4xCO₂ experiment is now explained in the manuscript as follows “we focus on understanding the proportionality in a 4xCO₂ scenario because atmospheric CO₂ and radiative forcing are held constant throughout the experiment, making the climate response easier to represent analytically”.

The reviewer's second question is whether the exponentially decaying solution I find for $q(t)$ and $f(t)$ is the only possible solution. I agree that this point was not entirely clear in the original manuscript. Therefore I have substantially expanded the Analytical Model section to clarify this. The manuscript now first describes how a solution with atmosphere-land flux consisting of a constant plus a term proportional to atmosphere-ocean flux is mathematically a possible solution of our equations. A constant flux is ruled out by the requirement that the fluxes must approach zero as the system approaches equilibrium in the 4xCO₂ experiment with constant atmospheric CO₂ concentration. Further, Solomon et al. (2009) argued that the proportionality of warming to emissions arose because atmosphere-ocean heat and atmosphere-ocean carbon fluxes were proportional. And in our experiments, atmospheric CO₂ concentration is constant, and therefore atmosphere-ocean carbon fluxes have no influence whatsoever on atmosphere-land carbon fluxes. Therefore, there is no physical basis for an atmosphere-land carbon flux being proportional to the atmosphere-ocean carbon flux, and therefore I set it to zero. Making this substitution in equation (5), it is now clear that the solutions I find for $q(t)$ and $f(t)$ are the only possible solutions, and I now explicitly show this in the manuscript.

3) Now, if I accept that it makes sense to compare the simple model as developed by the author to ESM results, I find the comparison is unnecessarily strict. All model parameters for the simple model seem to be taken from the respective ESM (according to the methods section). However, the purpose here is to test whether a model in which the proportionality of warming and cumulative emissions follows from $q(t) \sim f(t)$ could reproduce (partly) ESM results. The purpose is not to test whether a single layer EBM and

a very simple carbon cycle model with realistic parameters (=parameters corresponding to an ESM) can reproduce results from an ESM simulation (we know that it can't). So, it would be more appropriate to use one of the parameters to tune the simple model to the respective ESM. Since λ cancels out in the calculation of TCRE (eq. 9), this would be a possible choice. My suspicion is that through "tuning" of λ it will be possible to obtain a relatively good fit for Delta T and carbon fluxes for the last 100 years of the simulation in all ESMs (as is already the case for 2 of the models). If this would work, then the mismatch would come down to a question of timescales: For the short timescale the simple model doesn't explain ESM results, but for the longer it approximately can (this is already the case for MIP-ESM and NorESM2).

Thanks to the review for raising this question. The idea here is to see how realistic the main features are of our analytical model in which atmosphere-ocean heat and carbon fluxes are proportional, and in which warming is proportional to emissions, compared to the ESMs. The idea is not to individually fit each ESM variable as accurately as possible – for example, even if the simple model parameters could be adjusted to perfectly match the evolution of global mean temperature, this comparison would still not support the validity of the Solomon et al. hypothesis since the ESMs do not exhibit proportional heat and carbon fluxes. Nonetheless, I do compare the evolution of each variable in the simple model with that in the ESMs, and thus I discuss the reviewer's suggestions for improving the match below.

I first consider the reviewer's suggestion to vary the climate feedback parameter to better fit ESM results. First note that (by considering equation (6) at equilibrium):

$$\frac{F_{4 \times CO_2}}{\lambda} = 2 \times ECS$$

As noted in the methods section, I use this formula to calculate λ for each model from its ECS. Also, note that ECS for each model was estimated directly from its 4xCO₂ experiment by Schlund et al. (2020), which I reference, and I use their values. In my equations for heat and carbon fluxes, and temperature and cumulative emissions, λ always appears as a ratio with $F_{4 \times CO_2}$, a ratio which I could replace $2 \times ECS$. Thus the proposal to adjust λ to give a better fit over the 150 year period shown in Figure 2 is equivalent to adjusting each model's ECS to give a better fit over this period. For physical consistency I prefer to retain the values of ECS for each model which were previously diagnosed from these same experiments and reported in the literature.

In principle, I could adjust f_0 , so that instead of the atmosphere-ocean carbon flux agreeing with the ESM at time $t=0$, it gave agreement with the ESM's atmosphere-ocean carbon at some later time. Doing this would reduce the simple models' atmosphere-ocean carbon flux, and increase the e-folding timescale for all quantities. This would give better agreement of temperature, cumulative emissions and heat fluxes on long timescales, at the expense of worse agreement on shorter timescales. But fundamentally, the simple model would still fail to capture key features of the ESM response, in particular it would still assume that the cumulative emissions are due entirely to ocean carbon uptake, and that heat and carbon fluxes are proportional, which are far from true in the ESMs. And choosing a time to match the ESM carbon flux would be somewhat arbitrary. Therefore I prefer to retain the approach of fitting the atmosphere-ocean carbon flux at $t = 0$.

I disagree that the hypothesis of proportional warming to cumulative emissions because of proportional heat and carbon fluxes is consistent with the ESMs on certain timescales. As shown in Figure 3, there is no range of timescales for which the ratio of heat to carbon fluxes is constant in all models, and there is no range of timescales over which ocean carbon uptake dominates cumulative emissions. While it is true that I might be able to fit the temperature response over the early or late part of the experiment better by adjusting the value of ECS for each model, I would still not recover these key features of the Solomon et al. hypothesis.

Other points:

In the 2nd paragraph of the text, the author writes: "As shown in Figure 1, this property implies that if warming is proportional to cumulative emissions under one type of emissions scenario, it will be proportional to warming in all others." In my opinion, Fig. 1 has little explanatory power. I would rather prefer some (mathematically based) inference, or a reference in which such can be found.

Thanks to the reviewer for the suggestion. In response to this comment I have added a mathematical explanation, in equations 1-5, and associated text, for why the Caldeira and Kasting result implies that if warming is proportional to cumulative emissions in one scenario, it will also be proportional in all others. I have retained Figure 1 to provide a visual illustration of this.

In the derivation of the simple model it is assumed early on that $\Delta C_l(t)=0$ (stating that this is done because IPCC has assessed that it is not important). But doesn't $\Delta C_l(t)=0$ also follows from the derivation of the model itself, if one imposes to have $\Delta T \sim E$ (same as for ΔC_a). So, wouldn't it make more sense to still have ΔC_l in eq. (8) and then state that it needs to be zero for deriving eq. (9)?

Thanks to the reviewer for raising this question. I have revised this section of the manuscript substantially in response to this comment, and no longer state that we assume $\Delta C_L(t) = 0$ based on the IPCC assessment. The revised manuscript now discusses how atmosphere-land carbon flux, $\frac{d\Delta C_L(t)}{dt}$, with a component proportional to $q(t)$, and a constant component, is a possible solution of equation 10. However, as equilibrium is approached in the 4xCO₂ experiment, with its constant atmospheric CO₂ concentration, the fluxes must approach zero, and therefore the constant component of the flux must be zero. The text also now explains

"the atmosphere-ocean carbon flux does not influence the atmosphere-land carbon flux at all in the experiment considered, since the atmospheric concentration of CO₂ is constant, hence there is no physical reason why the atmosphere-land carbon flux should be proportional to the atmosphere-ocean carbon flux. Hence, if the Solomon et al. hypothesis is valid, and the proportionality between warming and cumulative emissions is driven by the proportionality between heat and carbon fluxes into the ocean, then land uptake of carbon must be negligible".

The author states several times that the results obtained with the simple model contradict the study of

Solomon et al. (2009). It is worth noting, however, that Solomon et al. consider the climate system after complete cessation of all emissions and on long time scales (1000 years), something that is not at all represented by the model experiment investigated here. So, I would recommend a more careful wording here.

Thanks to the reviewer for raising this point. When I introduce the Solomon et al. hypothesis I do state that Solomon et al. were considering the response after a complete cessation of emissions, and I directly quote from Solomon et al., including the reference to the long-term response.

“Solomon et al. argue that the proportionality of warming to emissions following a complete cessation of emissions ‘arises because long-term carbon dioxide removal and ocean heat uptake are both dependent on the same physics of deep-ocean mixing’, and subsequent studies provide a similar explanation”.

In response to this and other comments, I do also now show and discuss how my simple model based on the Solomon et al. hypothesis also does not provide a good fit to simulations of the response following a complete cessation of emissions (Supplementary Figures 1 and 2), which is the scenario that Solomon et al. were considering. Nonetheless, I accept the reviewer’s point that the hypothesis I am testing, that warming is proportional to emissions because atmosphere-ocean heat and carbon fluxes are proportional on all timescales, does go beyond what Solomon et al. say. Therefore, in the Discussion, where I focus on the implications of my findings, I have moderated the language, replacing ‘the Solomon et al. hypothesis’ with ‘the hypothesis that warming is proportional to emissions because atmosphere-ocean heat and carbon fluxes are proportional’, and the second instance of ‘the Solomon et al. hypothesis’ with ‘this hypothesis’.

REVIEWERS' COMMENTS

Reviewer #1 (Remarks to the Author):

This revised manuscript has addressed all comments arising from the first version. The results are certainly of relevance to the community and are an important and significant novel contribution to the literature.

The methods support the results and conclusions, and the work is placed nicely within the context of the existing literature. The work can be easily reproduced from the material.

A nice conceptual mathematical approach is used to gain understanding and insight into a widely observed phenomena (the near proportionality of warming to emissions across timescales). The methods also includes comparison of this conceptual mathematical framework with complex CMIP6 climate output. The findings will likely be widely cited within the literature.

For the above reasons, I support publication of the manuscript in its present form.

Reviewer #2 (Remarks to the Author):

I find that in the revised manuscript "Is warming proportional to cumulative carbon emissions because heat and carbon share ocean mixing processes" by Nathan Gillett, the author has adequately addressed my concerns from the review of the first version of this manuscript. I have a few remaining comments, and I would recommend this manuscript for publication after these points have been addressed.

I have to come back to the time-scale of the Solomon et al. hypothesis: The author quotes the hypothesis ('arises because long-term carbon dioxide removal and ocean heat uptake are both dependent on the same physics of deep-ocean mixing'), so this hypothesis was formulated for the 'long-term' and invokes 'deep-ocean mixing' as the mechanism. 'Long term' has to be seen in the context of the 1200 years of model simulations used by Solomon et al. In contrast, the Author uses 150 year long CMIP6 4xCO₂ simulations, i.e. the timescale considered in the manuscript is much shorter. This might be an important difference, since, for example, land carbon fluxes tend to

weaken substantially (or even reverse) in the longer term in ESMs (see e.g. Koven et al. 2022, <https://doi.org/10.5194/esd-13-885-2022>) and atmosphere changes also become much slower after the initial strong transients have ceased. Also, the process of deep-ocean mixing is certainly very different from shallow penetration of heat and carbon into the mixed layer that probably dominates the short timescales. In this sense, I do not believe that the manuscript says much about the original Solomon et al. hypothesis simply because of the different time-scale (I acknowledge that Solomon et al. didn't prove the validity of their hypothesis for their simulations either). The strength of the manuscript is to show that this hypothesis cannot simply be transferred to shorter timescales and transient climate states, and I think the manuscript would benefit from making this more explicit.

For example, in the 2nd paragraph it says 'Here we focus on the proportionality across a range of timescales, ...'. Why not say "... decadal to centennial time scales"?

In the first paragraph of the Discussion: I would suggest to highlight that the original Solomon hypothesis was echoed in subsequent publications also for short time-scales for which it wasn't intended, and the author shows that it doesn't hold for these shorter time scales.

Last paragraph of the Discussion could say "Overall, on the decadal to centennial time scales considered here, ..."

Minor points:

Introduction, first paragraph: (AR6) -> (AR5 and AR6)

Introduction, second paragraph: '... requires two distinct properties of the climate system. The first is...' In the revised text a second property is never explicitly mentioned anymore, which is a bit confusing.

Equation 3: Why $E(t-t'')$? Shouldn't this just be $E(t')$?

Finally, please acknowledge the ESM modelling groups and WCRP as required in the CMIP6 terms of use (<https://pcmdi.llnl.gov/CMIP6/TermsOfUse/TermsOfUse6-2.html>).

Response to reviewers' comments on revised version of "Is warming proportional to cumulative carbon emissions because heat and carbon share ocean mixing processes?"

Reviewer #1 (Remarks to the Author):

This revised manuscript has addressed all comments arising from the first version. The results are certainly of relevance to the community and are an important and significant novel contribution to the literature.

The methods support the results and conclusions, and the work is placed nicely within the context of the existing literature. The work can be easily reproduced from the material.

A nice conceptual mathematical approach is used to gain understanding and insight into a widely observed phenomena (the near proportionality of warming to emissions across timescales). The methods also includes comparison of this conceptual mathematical framework with complex CMIP6 climate output. The findings will likely be widely cited within the literature.

For the above reasons, I support publication of the manuscript in its present form.

Thanks to the review for the positive comments on the manuscript, and I'm pleased that he/she now supports publication.

Reviewer #2 (Remarks to the Author):

I find that in the revised manuscript "Is warming proportional to cumulative carbon emissions because heat and carbon share ocean mixing processes" by Nathan Gillett, the author has adequately addressed my concerns from the review of the first version of this manuscript. I have a few remaining comments, and I would recommend this manuscript for publication after these points have been addressed.

I have to come back to the time-scale of the Solomon et al. hypothesis: The author quotes the hypothesis ('arises because long-term carbon dioxide removal and ocean heat uptake are both dependent on the same physics of deep-ocean mixing'), so this hypothesis was formulated for the 'long-term' and invokes 'deep-ocean mixing' as the mechanism. 'Long term' has to be seen in the context of the 1200 years of model simulations used by Solomon et al. In contrast, the Author uses 150 year long CMIP6 4xCO2 simulations, i.e. the timescale considered in the manuscript is much shorter. This might be an important difference, since, for example, land carbon fluxes tend to weaken substantially (or even reverse) in the longer term in ESMs (see e.g. Koven et al. 2022, <https://doi.org/10.5194/esd-13-885-2022>) and atmosphere changes also become much slower after the initial strong transients have ceased. Also, the process of deep-ocean mixing is certainly very different from shallow penetration of heat and carbon into the mixed layer that probably dominates the short timescales. In this sense, I do not believe that the manuscript says much about the original Salomon et al. hypothesis simply because of the different time-scale (I acknowledge that Solomon et al. didn't prove the validity of their hypothesis for their simulations either). The strength of the manuscript is to show that this hypothesis cannot simply be transferred to shorter timescales and transient climate states, and I think the manuscript would

benefit from making this more explicit.

For example, in the 2nd paragraph it says 'Here we focus on the proportionality across a range of timescales, ...'. Why not say "... decadal to centennial time scales"?

Suggested change made.

In the first paragraph of the Discussion: I would suggest to highlight that the original Solomon hypothesis was echoed in subsequent publications also for short time-scales for which it wasn't intended, and the author shows that it doesn't hold for these shorter time scales.

Thanks for the comment. I have revised the discussion in the two ways suggested to make the point that the Solomon et al. hypothesis was focused on long timescales, that other publications use the hypothesis to explain the proportionality over a range of timescales, and (through the change made to the final paragraph) that this paper demonstrates that this hypothesis is invalid on decadal to centennial timescales. First paragraph of discussion revised to read "Solomon et al.¹⁵ argued that global warming is proportional to cumulative carbon emissions because long-term heat and carbon removal by ~~are mixed into~~ the ocean are driven by the same physical processes, and a similar argument has been used to explain the proportionality across a range of timescales echoed in many subsequent publications^{2,16,17}, including in the IPCC AR6⁴."

In making these revisions, I also noticed another statement in IPCC AR6 Chapter 5 on the near-linear relationship between warming and cumulative emissions ('the near-linear relationship between cumulative CO₂ emissions and global warming (TCRE) is thought to arise, to a large extent, from the compensation between the decreasing ability of the ocean to take up heat and CO₂ at higher cumulative CO₂ emissions, pointing to similar processes that determine ocean uptake of heat and carbon') and have added this to the final paragraph of the introduction. I also noticed that the IPCC AR6 Chapter 5 statement on the role of the land carbon sink ('the land carbon sink does not appear to play an important role in determining the linearity and path-independence of TCRE') was missing and now quote this in the results section.

Last paragraph of the Discussion could say "Overall, on the decadal to centennial time scales considered here, ..."

Suggested change made.

Minor points:

Introduction, first paragraph: (AR6) -> (AR5 and AR6)

Suggested change made.

Introduction, second paragraph: '... requires two distinct properties of the climate system. The first is...'
In the revised text a second property is never explicitly mentioned anymore, which is a bit confusing.

Thanks to the reviewer for highlighting this omission. I have added this sentence to the third paragraph of the introduction to explain what the second required property is: ‘Hence the second property of the climate system needed for proportionality of warming to emissions in general, is warming proportional to cumulative emissions in a $4\times\text{CO}_2$ experiment.’

Equation 3: Why $E(t-t'')$? Shouldn't this just be $E(t'')$?

I could replace $E(t-t'')$ with $E(t'')$ in equation (3), but this would require replacing $w_2(t'')$ with $w_2(t-t'')$, or redefining w_2 . I prefer to keep the equation as it is.

Finally, please acknowledge the ESM modelling groups and WCRP as required in the CMIP6 terms of use (<https://pcmdi.llnl.gov/CMIP6/TermsOfUse/TermsOfUse6-2.html>).

Following the terms of use, I have added the following statement to the acknowledgements: ‘I acknowledge the World Climate Research Programme, which, through its Working Group on Coupled Modelling, coordinated and promoted CMIP6. I thank the climate modeling groups for producing and making available their model output, the Earth System Grid Federation (ESGF) for archiving the data and providing access, and the multiple funding agencies who support CMIP6 and ESGF.’